

# Analysis of ceRNA network of differentially expressed genes in FaDu cell line and a cisplatin-resistant line derived from it

Gehou Zhang, Guolin Tan, Tieqi Li, Jingang Ai, Yexun Song, Zheng Zhou, Jian Xiao and Wei Li

Department of Otolaryngology-Head Neck Surgery, Third Xiangya Hospital of Central South University, Changsha, Hunan, China

## ABSTRACT

**Background:** Hypopharyngeal cancer accounts for 2% in head and neck cancers and has a poor prognosis. Cisplatin is a widely used chemotherapeutic drug in kinds of carcinomas, concluding hypopharyngeal cancer. However, the resistance of cisplatin appeared in recent years. Cisplatin-resistance has been partly explored before, but rarely in hypopharyngeal cancer.

**Methods:** We cultured the hypopharyngeal cancer cell (FaDu) and induced its cisplatin-resistant cell (FaDu/DDP4). Then we tested the differentially expressed genes (DEGs) between FaDu and FaDu/DDP4. Gene Ontology (GO) and Kyoto Encyclopedia of Genes and Genomes (KEGG) analyses were conducted on the DEGs, and we drew the ceRNA networks of DEGs. Finally, we chose eight miRNAs and six mRNAs for qRT-PCR to verify our microarray.

**Results:** We induced cisplatin-resistant FaDu/DDP4 and proved its chemoresistance. The resistance index (RI) of FaDu/DDP4 was 2.828. DEGs contain 2,388 lncRNAs, 1,932 circRNAs, 745 mRNAs and 202 miRNAs. These 745 mRNAs were classified into three domains and 47 secondary GO terms. In KEGG pathway enrichment, the "TNF signaling pathway", "IL-17 signaling pathway" and "JAK-STAT signaling pathway" were potentially significant signaling pathways. Then, 52 lncRNAs, 148 circRNAs, 155 mRNAs and 18 miRNAs were selected to draw the network. We noticed several potential targets (as *miR-197-5p*, *miR-6808-5p*, *APOE*, *MMP1*, *S100A9* and *CYP24A1*). At last, the eight miRNAs and six mRNAs that are critical RNAs in ceRNA network were verified by qRT-PCR.

**Conclusion:** The microarray helped to find DEGs in cisplatin-resistant hypopharyngeal cancer. TNF, IL-17 and JAK-STAT signaling pathways might be more significant for cisplatin-resistance. *MiR-197-5p*, *miR-6808-5p*, *APOE*, *MMP1*, *S100A9* and *CYP24A1* might be potential genes inducing resistance.

Corresponding author
Wei Li, lilywei1979@126.com

## INTRODUCTION

Hypopharyngeal cancer is a rare kind of cancer that occurs in the laryngopharynx. It accounts for 0.15% to 0.24% in systematic malignant tumors and 2% in head and neck cancers (*Subspecialty Group of Head & Neck Surgery, 2017*). Hypopharyngeal cancer has

the same etiologic factors with laryngeal cancer. The causes contain alcohol, tobacco, chewing areca nut and others (*Rahman et al., 2020*). But hypopharyngeal cancer has poorer prognoses because of its initially silent development (*Rahman et al., 2020*). Patients in terminal stage and before surgery or radiotherapy always receive induction chemotherapy. The universal regimens are PF (cisplatin and 5-Fu) or TPF (pacitaxel, cisplatin and 5-Fu). However, because of the massive use in malignant tumors, acquired cisplatin-resistance appears. Malignant cells activate kinds of adaptive responses and form chemoresistance to cisplatin (*Galluzzi et al., 2014*). Due to the drug-resistance, the clinical utility of cisplatin is limited in some degree (*Amable, 2016*).

LncRNA is the abbreviation of long non-coding RNA (all abbreviations in this article were listed in Table S1). They have more than 200 nucleotides and do not encode for proteins (*Fatica & Bozzoni, 2014*). But they regulate genes by targeting chromatin regulators involved in gene silencing (*Rinn & Chang, 2012*). Further researches found that lncRNA regulates gene expression through cis- or trans-regulatory mechanisms. LncRNA exerts cis-regulation on an adjacent gene on the same allele (*Guttman & Rinn, 2012*). Moreover, lncRNA and mRNA can competitively combine miRNA. LncRNA can sponge miRNA to indirectly regulate mRNA expression (*Paraskevopoulou et al., 2013*). Many studies have demonstrated that lncRNAs can target miRNAs and regulate chemoresistance. For example, lncRNA *KCNQ1OT1* facilitated cisplatin-resistance in tongue cancer by targeting *miR-221-5p* (*Zhang et al., 2018b*).

CircRNA has a covalently closed loop and contains one or more exon sequences (*Chen & Yang, 2015*). CircRNA can also serve as a 'sponging' model that absorbs miRNAs and competes with mRNAs (*Wilusz & Sharp, 2013*). Recently, many studies have verified that their dysregulation influence chemoresistance (*Hua et al., 2019*). For example, *hsa_circ_001569* was up-regulated in drug-resistance osteosarcoma cell lines. This elevation enhanced cell proliferation and resistance to cisplatin (*Zhang et al., 2018a*).

MicroRNA is a kind of noncoding RNA approximately 22 nucleotides long (*Ambros, 2001*). They work as regulators of gene expression by targeting different regions of mRNAs. It's commonly known that miRNA combine the 3′UTR of mRNA in the cytoplasm to repress mRNA translation (*Bartel, 2004*). But a recent study indicated that miRNA can target enhancer and activate transcription in the nucleus (*Xiao et al., 2017*). No matter what mechanism is adopted, miRNA can undoubtedly regulate gene expression. The differential expression of diverse miRNAs has been verified in various cancers. The conception of competitive endogenous RNAs (ceRNAs) was proposed in recent 10 years. Researchers consider ceRNAs as a microRNA sponges that contain microRNA response elements. They competitively bind to microRNA as the microRNA silencing element (*Salmena et al., 2011*). Multiple non-coding RNAs, such as lncRNA, circRNA, small non-coding RNA and pseudogenes, can act as ceRNA (*Tay, Rinn & Pandolfi, 2014*). Due to their function, the axis of lncRNA/circRNA-miRNA-mRNA could be easily imagined. Several axes compose the ceRNA network. It can help to reveal the potential and significant biomarkers and therapy targets.

In this article, we cultured the FaDu cell, a cell line of hypopharyngeal cancer. And we induced its cisplatin-resistant cell, named as FaDu/DDP4 cell. We tested differentially expressed genes (DEGs) between the FaDu cell and the FaDu/DDP4 cell, and drew the ceRNA networks of DEGs.

## MATERIALS & METHODS

### Cell culture and chemoresistant cell line induction

The FaDu cell lines were purchased from Suzhou Bei Na Chuanglian Biotechnology Co., Ltd., China (#BNCC316798). FaDu cells were cultured with MEM medium containing 10% FBS in 25 cm culture flasks at 37 °C with 5% $CO_2$.

To induce cisplatin-resistant cell line, cisplatin was added into the culture medium. After cisplatin affecting for 24 h, the residual cells were cultured without cisplatin to form a new generation. The initial concentration of cisplatin was 1.67 μM to induce the first and second generation, while 3.3 μM for the next two generations. FaDu/DDP4 cells were formed when we induced the forth generation. FaDu/DDP4 cells were cultured in MEM medium containing low concentration of cisplatin (0.5 μM). Before its use for experiment, the medium was changed to cisplatin-free medium 2 weeks ago.

### Colony formation experiment

A total of $1 \times 10^3$ cells in 1ml MEM medium per well were seeded into a 24-well plate, 12 wells of FaDu cells and 12 wells of FaDu/DDP4 cells. After they were cultured for 24 hours, the FaDu cells and FaDu/DDP4 cells were added with different concentrations of cisplatin. The concentration gradient of cisplatin was 0, 0.83, 2.5 and 5 μM per well. After cisplatin affecting for 24 h, the medium was changed to cisplatin-free medium and cells were cultured for two weeks. Then, cells were fixed with Methanol and dyed with Giemsa stain. Image-Pro was used to count for colony numbers. Compared with no cisplatin wells, the proportion of decreased colonies in other wells was colony inhibition. The colony inhibition of 50% was the half inhibitory concentration (IC50). The ratio of IC50 of FaDu/DDP4 to IC50 of FaDu was resistance index (RI) of FaDu/DDP4.

### Gene microarray, GO and KEGG analysis

Each four 25 cm culture flasks of FaDu and FaDu/DDP4 were used to extract RNA. Total RNA was extracted and purified using miRNeasy Mini Kit (Cat#217004, QIAGEN, GmBH, Germany). Gene microarray was performed by Shanghai Biotechnology Co., Ltd., China.

GO (Gene Ontology) and KEGG (Kyoto Encyclopedia of Genes and Genomes) pathway were analyzed by online software DAVID (https://david.ncifcrf.gov/). GO classification classified DEGs into three domains containing 47 second level GO terms. These DEGs were enriched to over two hundred KEGG pathways. Pathways were arrayed in order of enrich factor. Enrich factor was defined as (the proportion of DEGs in a pathway)/ (the proportion of all DEGs). Pathways with $p$-value less than 0.05 were selected and then pathways absolutely irrelative to cancer cell were excluded.

## ceRNA networks analysis

We set a standard that fold-change(abs) is no less than 2 and normalized expression in FaDu or FaDu/DDP4 is no less than 7. (Fold-change(abs) is $2^{|\text{expression in FaDu/DDP4} - \text{expression in FaDu}|}$.) This standard was applied to select differential lncRNAs, circRNAs and mRNAs. Then, lncRNA/circRNA to miRNA and mRNA to miRNA matches were found according the RNA sequences through miRanda. MiRNAs in lncRNA/circRNA-miRNA matches intersected miRNAs in mRNA-miRNA matches. And then, miRNAs in the intersection were picked out if their fold-change(abs) were no less than 1.2. LncRNAs, circRNAs and mRNAs related to these miRNAs were used to draw the ceRNA network. All networks were drawn by Cytoscape version 3.6.0.

## Quantitative real-time PCR (qRT-PCR)

E.Z.N.A.® miRNA Kit was used to extract long strand RNA and miRNA from FaDu cells and FaDu/DDP4 cells. Concentration of RNA was tested by Thermo® NanoDrop 2000. MiRNAs was amplified by tailing reaction. All-in-One™ miRNA First-Strand cDNA Synthesis Kit and miRNA qRT-PCR Detection Kit were from GeneCopoeia Inc., America. ReverTra Ace® qPCR RT Master Mix with gDNA Remover and KOD SYBR® qPCR Mix (TOYOBO, Japan) were used for mRNA reverse transcription and qRT-PCR. Primers for miRNA were designed and synthesized by Shanghai Sangon Biotech, China (sequences of these primers were in Table S2). Primers for mRNAs were from GeneCopoeia Inc., USA. Roche LightCycler® 480 was used to detected amplification. The expression of miRNAs was normalized by *RNU6* and mRNAs by *GAPDH* and *ACTB*.

## Statistical analysis

All experiments were designed and performed with three technical replicates. All results are presented as mean ± SD. Unpaired Student's t-test calculated the statistical significance, performed with IBM® SPSS statistics 25. The figure of IC50 and qRT-PCR was drawn by GraphPad Prism 5. P-value less than 0.05 was considered statistically significant.

# RESULTS

## Chemoresistant cell line was induced and its chemoresistance was verified

We induce ciplatin-resistant FaDu cells through short cisplatin stimulating. Because of the drug, cell nuclei became bigger and vacuolated, and cells formed pseudopodia. After several cells died, the remaining cells recovered and grew up without cisplatin. These cells were regarded as a new generation of steadily chemoresistant cells. The fourth generation of chemoresistant FaDu cells was named as FaDu/DDP4 cells. Their morphology remains mildly different from FaDu cells. FaDu/DDP4 cells had slightly vacuolated, bigger cell nuclei and residual pseudopodia (Figs. 1A and 1B).

The colony formation experiment shows that FaDu/DDP4 have greater cisplatin-resistance than FaDu. At 0 μM cisplatin, the colony numbers of FaDu/DDP4 cell is almost

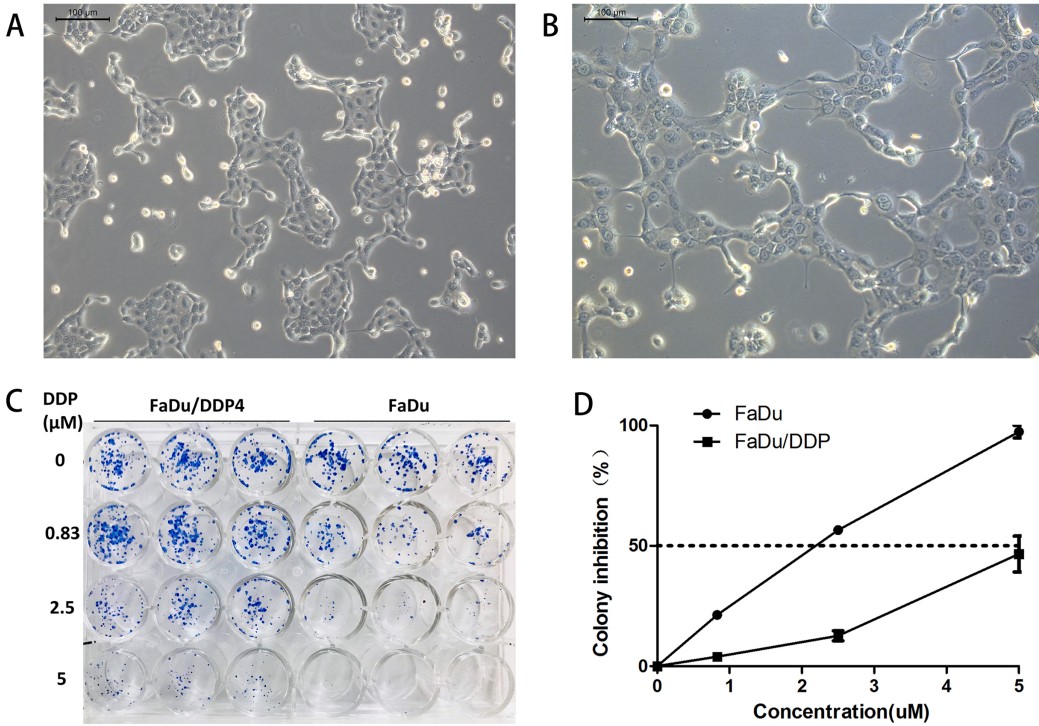

**Figure 1 Cell morphology comparison and colony formation experiment.** (A) The morpholoy of FaDu cells under microscope (10 × 10). (B) The morphology of FaDu/DDP4 cells under microscope (10 × 10). Cell nuclei are bigger and vacuolated, and pseudopodia can be seen. (C) The colony formation experiment. At 0 μM cisplatin, FaDu cells are almost equal to FaDu/DDP4 cells. At 0.83, 2.5 and 5 μM cisplatin, FaDu cells are fewer than FaDu/DDP4 cells. The chemoresistance of FaDu/DDP4 cells has been demonstrated. (D) Line chart of colony inhibition at different concentration of cisplatin. The dotted line means that colony inhibition is 50%. 

equal to FaDu cell colonies. But the colony numbers of FaDu cell are all more than FaDu cell colonies at 0.83, 2.5 and 5 μM cisplatin (Fig. 1C). We counted the colony numbers of each well to calculate the colony inhibition. 2.5 μM cisplatin inhibited over 50% FaDu cells, while less than 25% FaDu/DDP4 cells. A total of 5 μM cisplatin inhibited almost 100% FaDu cells but less than 50% FaDu/DDP4 (Fig. 1D). Besides, the calculated IC50 of cisplatin is 1.873 μM in FaDu and 5.296 μM in FaDu/DDP4. Thus, the resistance index (RI) of FaDu/DDP4 was 2.828.

## Gene microarray shows DEGs between two cell lines

The microarray totally tested 58,539 lncRNAs, 88,371 circRNAs, 18,263 mRNAs and 2,549 miRNAs. The fold-changes(abs) of differential expressed lncRNAs, circRNAs and mRNAs are no less than 2, and of miRNAs is no less than 1.2. Up-regulated RNAs are RNAs expressing higher in FaDu/DDP4, and down-regulated means lower. A total of 2,388 lncRNAs, 1,932 circRNAs, 745 mRNAs and 202 miRNAs were differential. A total of 1,234 lncRNAs, 1,016 circRNAs, 527 mRNAs and 191 miRNAs were up regulated, and 1,154 lncRNAs, 916 circRNAs, 218 mRNAs and 11 miRNAs were down regulated (Table 1).
**Table 1 Statistics of differentially expressed genes.**

|           | Totally tested | Differentially expressed | Up regulated | Down regulated |
| --------- | -------------- | ------------------------ | ------------ | -------------- |
| lncRNAs   | 58,539         | 2,388                    | 1,234        | 1,154          |
| circRNAs  | 88,371         | 1,932                    | 1,016        | 916            |
| mRNAs     | 18,263         | 745                      | 527          | 218            |
| miRNAs    | 2,549          | 202                      | 191          | 11             |

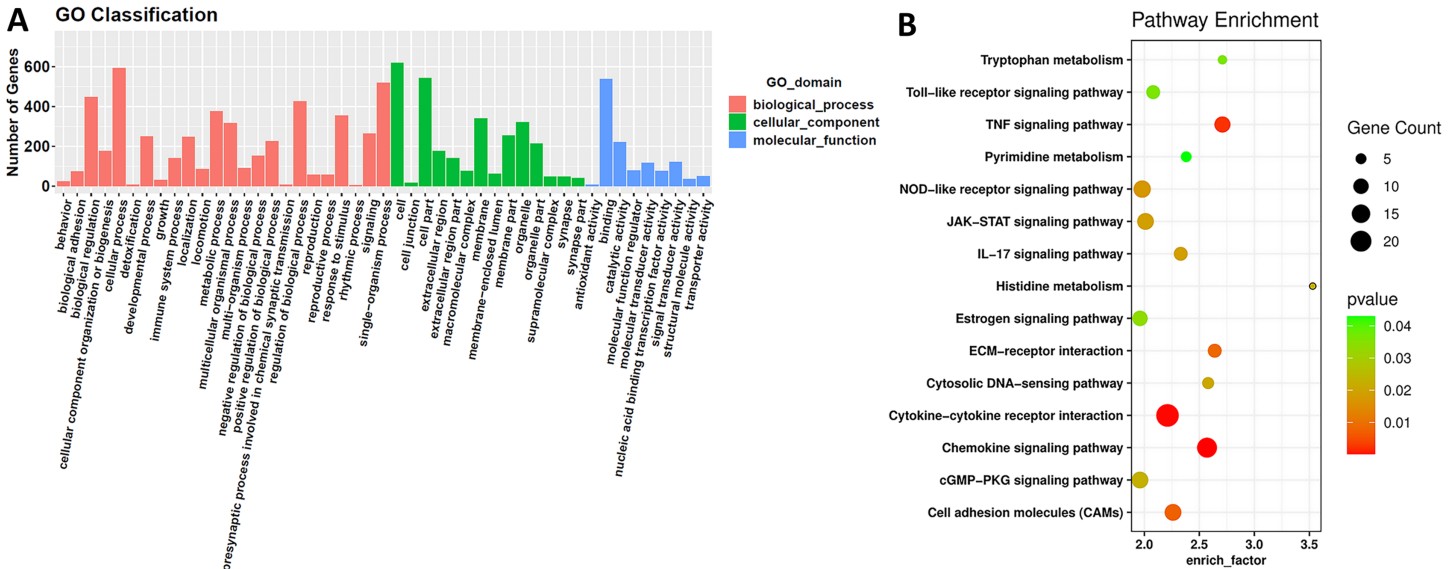

**Figure 2 GO and KEGG analysis of differentially expressed mRNAs.** GO and KEGG analysis. (A) GO classification. The differentially expressed mRNAs are classified into three initial GO domains and 47 second level of GO terms. The genes in each second level of GO term were counted and shown above. (B) KEGG analysis enriched differentially expressed mRNAs into 15 KEGG pathways with p-value less than 0.05 that may be involved in cancer progress and cisplatin-resistance.

### GO and KEGG analysis of differential expressed mRNAs

We classified differentially expressed mRNAs to different GO functions. GO classification divides differential expressed mRNAs into three domains and 47 second level GO terms (Fig. 2A). These three domains are biological process, cellular component and molecular function. Then, we enriched them to 278 KEGG pathways to understand their position in the pathways. After screened as the method discribed, 15 KEGG pathways were picked out and shown in Fig. 2B. These KEGG pathways concluded metabolism, cell adhesion and signal transmission. These biological processes might affect cisplatin-resistance.

### Function of lncRNAs and circRNAs as ceRNAs via lncRNA/circRNA-miRNA-mRNA networks

A total of 52 lncRNAs, 148 circRNAs, 155 mRNAs and 18 miRNAs were selected to draw the network diagrams (Table S3) (Fig. 3). The sizes of nodes in the networks are positively correlated to its interactive degree with other genes. These 18 miRNAs contain 1

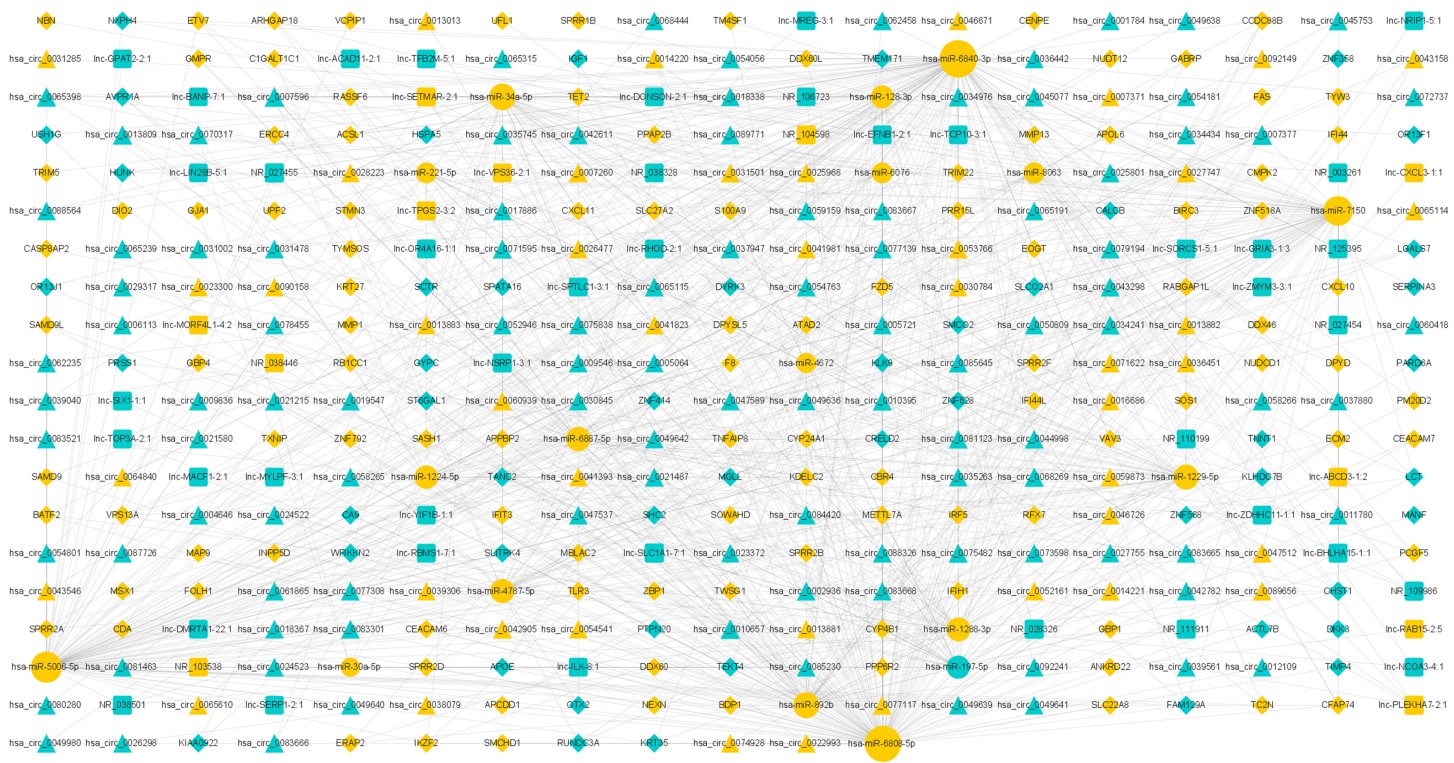

**Figure 3 ceRNA network of differentially expressed genes.** The ceRNA networks. Circles stand for miRNAs, triangles for circRNAs, squares for lncRNAs and diamonds for mRNAs. Blue genes are down regulated in FaDu/DDP4 cells compared to FaDu cells, and yellow RNAs are up regulated. Size of each gene is positively related to the degree, in which this gene interacts with other genes.

**Table 2 Fold-change and degree of miRNAs in ceRNA network.**

| miRNA_ID | Regulation | Foldchange | miRNA_ID | Regulation | Foldchange |
|---|---|---|---|---|---|
| hsa-miR-6840-3p | Up | 1.5226 | hsa-miR-1288-3p | Up | 1.5044 |
| hsa-miR-6808-5p | Up | 1.5403 | hsa-miR-128-3p | Up | 1.5023 |
| hsa-miR-5006-5p | Up | 1.5205 | hsa-miR-1224-5p | Up | 1.5579 |
| hsa-miR-7150 | Up | 1.5697 | hsa-miR-8063 | Up | 1.6654 |
| hsa-miR-34a-5p | Up | 1.8560 | hsa-miR-6076 | Up | 1.7099 |
| hsa-miR-1229-5p | Up | 1.9760 | hsa-miR-221-5p | Up | 1.5156 |
| hsa-miR-4787-5p | Up | 1.5729 | hsa-miR-4672 | Up | 1.7589 |
| Hsa-miR-892b | Up | 1.5260 | hsa-miR-30a-5p | Up | 1.7271 |
| hsa-miR-6887-5p | Up | 1.5915 | hsa-miR-197-5p | Down | 0.6213 |

down-regulated and 17 up-regulated miRNAs in FaDu/DDP4 cells (Table 2). Among all miRNAs in the ceRNA network above, we noticed *miR-197-5p* and *miR-6808-5p*. *MiR-197-5p* is the only down-regulated miRNA and *miR-6080-5p* has a bigger degree. The circRNAs, lncRNAs and mRNAs associated with *miR-197-5p* and *miR-6808-5p* were picked out. The other network centred on them was shown in Fig. 4.

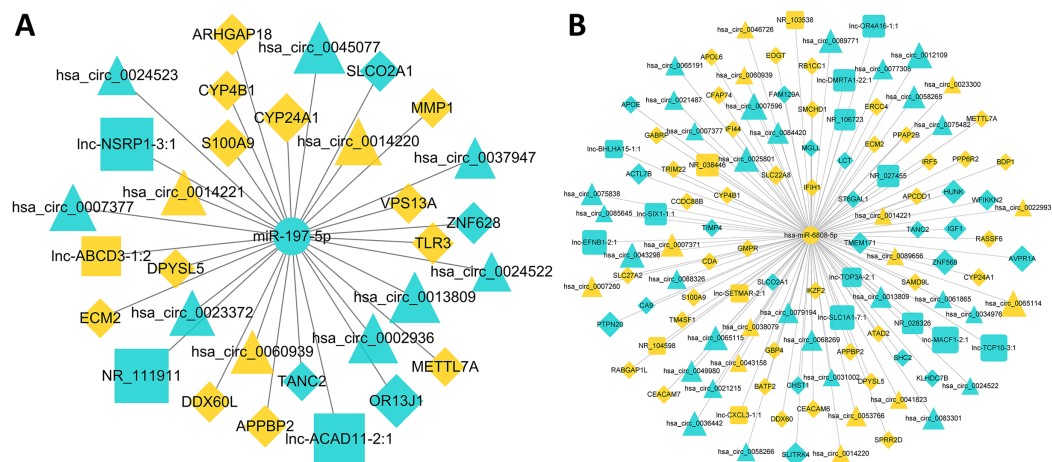

**Figure 4 ceRNA networks centred on *miR-197-5p* and *miR-6808-5p*.** Circle icons stand for miRNAs, triangular for circRNAs, quarate for lncRNAs and diamond for mRNAs. Blue means down regulated genes, while yellow means up regulated. The sizes of icons are positively correlated to fold-change(abs), standing for the expression difference between FaDu cells and FaDu/DDP4 cells. (A) The ceRNA network centred on *miR-197-5p*. (B) The ceRNA network centred on *miR-6808-5p*.

**Table 3 qRT-PCR result.**

| RNA Name | Fold-change | *p*-Value |
|---|---|---|
| *miR-197-5p* | 0.623 ± 0.054 | 0.0008 |
| *miR-6840-3p* | 1.450 ± 0.096 | 0.0031 |
| *miR-6808-5p* | 2.049 ± 0.117 | 0.0002 |
| *miR-5006-5p* | 1.841 ± 0.039 | <0.0001 |
| *miR-7150* | 1.467 ± 0.055 | 0.0018 |
| *miR-34a-5p* | 1.529 ± 0.044 | 0.0001 |
| *miR-892b* | 1.272 ± 0.075 | 0.0326 |
| *miR-122-5p* | 1.552 ± 0.056 | <0.0001 |
| *S100A9* | 19.700 ± 0.760 | <0.0001 |
| *MMP1* | 9.915 ± 0.119 | <0.0001 |
| *ARHGAP18* | 4.549 ± 0.470 | 0.0002 |
| *CYP24A1* | 4.439 ± 1.348 | 0.0116 |
| *APPBP2* | 2.169 ± 0.427 | 0.0101 |
| *APOE* | 0.393 ± 0.160 | 0.0161 |

**Note:**

Fold-changes and *p*-value of miRNAs and mRNAs. Fold-changes are shown as mean ± SD and *p*-values less than 0.05 were statistically significant.

## qRT-PCR verified the consequence of gene microarray

We chose eight miRNAs and six mRNAs in our ceRNA network for qRT-PCR to verify the consequence of gene microarray. The fold-changes and p-values of these RNAs were shown in Table 3 and Fig. 5. The result of qRT-PCR was statistically significant and consistent with our microarray.

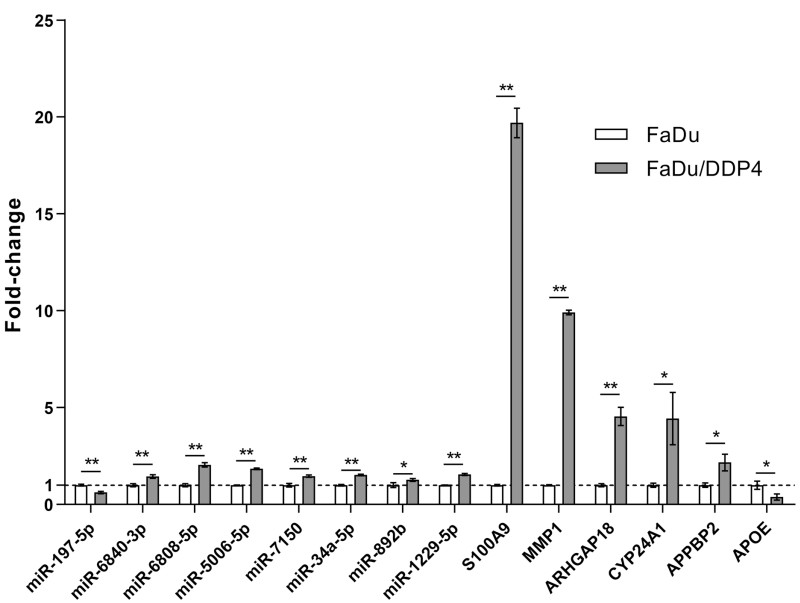

**Figure 5 qRT-PCR verifying microarray.** qRT-PCR. Eight miRNAs and six mRNAs were selected for qRT-PCR. Each gens was performed for three independent experiments. The results were shown as mean ± SD. *, $p < 0.05$; **, $p < 0.01$.

## DISCUSSION

Symptoms of hypopharyngeal cancer usually appear late, compared with other HNCs (*Rahman et al., 2020*). The patients of hypopharyngeal cancer are always diagnosed at a poorer stage and have a poorer prognosis (*Cooper et al., 2009*). Cisplatin is a common drug for induction chemotherapy of hypopharyngeal cancer. Thus, we focus on DEGs between cisplatin-resistant hypopharyngeal cancer cell line and its parent cell line.

We found that the cell nuclei became bigger and vacuolated, and pseudopodia appeared when cisplatin added into the medium. Moreover, the stable FaDu/DDP4 cells remained bigger nuclei and pseudopodia. It seems that the appearance of pseudopodia is a significant change in FaDu/DDP4 cells. We speculated that pseudopodia might related to epithelial-mesenchymal transition (EMT) or signal transmission. A research found that the cisplatin-resistant nasopharyngeal cancer (NPC) cells developed pseudopodia. Besides, EMT-related transcription factors (*Snail*, *Slug*, *Twist* and *ZEB1*) increased in them (*Zhang et al., 2014*). Similarly, the post-irradiation residual NPC cells also had pseudopodia. Their EMT related proteins were up regulated (*Su et al., 2016*). These findings might demostrate our speculation that pseudopodia are related to EMT.

Then, we used GO and KEGG analysis to classify and enrich differential mRNAs. In KEGG pathway enrichment (*Kanehisa et al., 2017*), we found 15 pathways that might participate in cisplatin resistance, shown in Fig. 2. Except for material metabolism pathways, we noticed some typical signaling pathways. "TNF signaling pathway" involves apoptosis, cell survival, regulation of transcription and others (Fig. S1) (*Kanehisa & Goto, 2000*). "IL-17 signaling pathway" also participates in immune process. This pathway relates to MAPK and NF-kB signaling pathway, which are common pathways in cancer.

And its downstream contains *S100A9* and *MMP1*, which are components in ceRNA network centered on *miR-197-5p* (Fig. S2). "JAK-STAT signaling pathway" influences cell cycle, survival and apoptosis. It also acts on MAPK and PI3K-AKT signaling pathway to take part in cancer processes (Fig. S3). Besides, other KEGG pathways reveals that extracellular matrix (ECM), cell membrane receptors and adhesion molecules were important in cisplatin-resistance.

We try to find the potential genes and their upstream non-coding RNAs inducing chemoresistance in vitro. So we draw the ceRNA network according to the result of our microarray. In the ceRNA network, we can see several lncRNA/circRNA-miRNA-mRNA axes. In the center, several miRNAs have been reported to serve as cancer-associated genes. In our microarray, *miR-128-3p* is up regulated in FaDu/DDP4, and might lead to chemoresistance. Cai et al. (2017) said that *miR-128-3p* activated Wnt/β-catenin and TGF-β pathways to induce EMT and chemoresistance in vitro, and it conferred tumorigenicity and metastasis in vivo. *MiR-197-5p* was down regulated according to our result. *MiR-197-5p* was thought to depress the expression of *E2F1*, which is a widely accepted oncogene, and it decelerated the glioma carcinogenesis (Li, Zhang & Wu, 2019). The high expressed *miR-221-5p* was positively correlate to poor prognosis of renal cell cancer (RCC), and *miR-221-5p* promotes the proliferation and migration of RCC cells (Liu et al., 2019). In prostate cancer cells, *miR-221-5p* was also demonstrated to enhance the proliferation, metastasis and EMT in vitro and in vivo (Shao et al., 2018). But in gastric cancer cells, *miR-221-5p* is low expressed. The overexpression of *miR-221-5p* reduces the cisplatin-resistance, proliferation and invasion(Jiang et al., 2020). Due to the controversial results of present studies, we can understand the complexity of miRNA function. How miRNAs affect the cisplatin-resistance of hypopharyngeal cancer needs deeper explorations.

In the ceRNA network focused on *miR-197-5p*, we found several genes participated in multiple tumor process. *APPBP2* encodes protein that interacts with microtubules. It is associated with beta-amyloid precursor protein (APP) transport or processing. APP relates to cancer cell viability, proliferation, migration and invasion. Silenced *APPBP2* inhibited cell proliferation, migration and aggressiveness and enhanced apoptosis in non-small cell lung cancer (Gong et al., 2019). *ARHGAP18*, Rho GTPase activated protein, is a member of RhoGAP family. Downregulation of *ARHGAP18* inhibited hepatocellular cancer (HCC) cell migration (Chen et al., 2018). Its knockdown reduced growth, migration and metastasis in triple negative breast cancer (Humphries et al., 2017). *CYP24A1*, a member of the cytochrome P450 superfamily of enzymes, encodes for 24-hydroxylase. This protein inactivates biologically active form of vitamin D (Jones, Prosser & Kaufmann, 2012). Overexpression of *CYP24A1* accelerated cell growth and cell invasion of lung adenocarcinoma cells. It also upregulated *RAS* in lung adenocarcinoma (Shiratsuchi et al., 2017). S100 calcium-binding protein A9 (*S100A9*) is reported to increase in several cancers. *S100A9* promoted cell proliferation and migration through activating the EMT and Wnt/β-catenin pathway in cervical cancer (Zha et al., 2019). Overexpressed *S100A9* decreased cisplatin sensitivity and apoptosis rate in squamous cervical cancer cells. This may be due to *S100A9* acting on the AKT/ERK-FOXO1-Nanog

signaling pathway (*Zhao et al., 2018*). Matrix metalloproteinase (MMP) is an important enzyme family, which contain mental ions and affect on ECM. Cancer-associated fibroblasts (CAFs) can produce MMPs. They can degrade ECM and facilitate cancer cell invasion and migration (*Das & Law, 2018*). *MMP1* promoted cell proliferation and migration of esophageal squamous cell carcinoma. This was attribute to activating the PI3K/AKT pathway (*Liu et al., 2016*). In the ceRNA network in *miR-6808-5p*, apolipoprotein E (*APOE*) was downregulated. Cancer-secreted APOE was regard as an anti-angiogenic and metastasis-suppressive factor. It engaged *LRP1* and *LRP8* receptors to repress invasion and metastatic endothelial recruitment (*Pencheva et al., 2012*). *APOE* could be targeted by *LXR* and suppress survival of myeloid-derived suppressor cells. Thus, *APOE* activated immunity in the tumor microenvironment (*Tavazoie et al., 2018*). Monoglyceride lipase (*MGLL*) was also down regulated. Overexpression of *MGLL* in HCC cells suppressed cell migration (*Yang et al., 2018*). *MGLL* overexpression inhibited AKT activation and cell proliferation (*Rajasekaran et al., 2016*). However, some other genes in the networks haven't been deeply explored in the cancer field. Therefore, the relationship between cisplatin-resistance and these DEGs needs our following research.

In our present study, we noticed several molecules and pathways inducing cisplatin-resistance. However, these results were based on bioinformatic analysis. The specific mechanisms need to be explored in the following experiments. Moreover, we only focused on cisplatin-resistance in hypopharyngeal cancer cell line. In further studies, we may integrate these results into other kinds of cisplatin-resistant cancers. And these results should be verified in vivo in the future.

## CONCLUSIONS

In this study, we induced a cisplatin-resistant hypopharyngeal cancer cell line. A total of 2,388 lncRNAs, 1,932 circRNAs, 745 mRNAs and 202 miRNAs were differentially expressed in Gene microarray. TNF, IL17 and JAK-STAT signaling pathways were potential signaling pathway participated in cisplatin-resistance. Genes centered on *MiR-197-5p* and *miR-6808-5p* in the network might be key genes inducing cisplatin-resistance.

### Funding

This work was supported by the National Natural Science Foundation of China (Nos. 81502358 and 81902785). The funders had no role in study design, data collection and analysis, decision to publish, or preparation of the manuscript.

### Grant Disclosures

The following grant information was disclosed by the authors:
National Natural Science Foundation of China: 81502358 and 81902785.

## Competing Interests

The authors declare that they have no competing interests.

## Author Contributions

- Gehou Zhang conceived and designed the experiments, performed the experiments, analyzed the data, prepared figures and/or tables, authored or reviewed drafts of the paper, and approved the final draft.
- Guolin Tan conceived and designed the experiments, authored or reviewed drafts of the paper, and approved the final draft.
- Tieqi Li performed the experiments, prepared figures and/or tables, and approved the final draft.
- Jingang Ai analyzed the data, authored or reviewed drafts of the paper, and approved the final draft.
- Yexun Song conceived and designed the experiments, performed the experiments, prepared figures and/or tables, and approved the final draft.
- Zheng Zhou analyzed the data, prepared figures and/or tables, and approved the final draft.
- Jian Xiao analyzed the data, authored or reviewed drafts of the paper, and approved the final draft.
- Wei Li conceived and designed the experiments, performed the experiments, analyzed the data, prepared figures and/or tables, authored or reviewed drafts of the paper, and approved the final draft.

## Microarray Data Deposition

The following information was supplied regarding the deposition of microarray data:

The microarray data are available at GEO. The GEO accession is GSE167399, containing two subseries GSE167396 and GSE167398.

The differentially expressed genes are available in the Supplemental File.

## Data Availability

The raw data are available in the Supplemental Files.

## Supplemental Information

Supplemental information for this article can be found online at http://dx.doi.org/10.7717/peerj.11645#supplemental-information.

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
