# Peer review of "Analysis of ceRNA network of differentially expressed genes in FaDu cell line and a cisplatin-resistant line derived from it"

_PeerJ, doi:10.7717/peerj.11645_

## Round 0.1 · original submission · Major Revisions

The manuscript was examined by two reviewers, both of whom have asked for a major revision. Please address each of their comments in the revision and/or your rebuttal document.

Reviewer 1 ·

Basic reporting

-English style needs a bit of improvement. There are a few typos in the following lines: 30-csplatin, 47-cord, 119 sift.
-Line 94, use script for 103.
-Scale bar missing in Fig 1A and 1B. Please include those!
-Some names in Fig 3 and 4B are overlapping, hence rendering them unreadable.
Please redo these figures.

Experimental design

At the moment, there is no information on how many biological replicates each experiment has. Please provide 'n' for each experiment.

Validity of the findings

-Fig 1D and 5 have the error bars. From how many experiments these data are represented? Are these the technical replicates or the biological replicates and how many experiments?
-For the statistics in Materials and Methods, the authors did not mention whether they used paired t-test or unpaired t-test. Please indicate that.

Additional comments

-For the qPCR results in line 186, the representation of the results in a table format is recommended. 
-In the discussion, please mention the limitations and future directions for this study.
-For Fig 2B, the indicated p-value range is from 0.1 to 0.5. Why these p-values are so large? What are they representing?

Reviewer 2 ·

Basic reporting

The title, abstract, introduction, methods, results and discussion are appropriate for the content of the text. Furthermore, the article is well constructed, the experiments are well conducted, and analysis is well performed. The figures are relevant, high quality, well labelled and described.

Experimental design

The experimental design is original and the research is within the scope of the journal.. Research question is well defined, relevant and meaningful. The methods are highly technical, ethical and logistical. Statistical methods are chosen correctly.

Validity of the findings

All underlying data have been provided in detail. The findings are meaningful. The conclusions are well stated and relevant to the research questions.

Additional comments

This paper investigates the function of expression of competitive endogenous RNAs (ceRNAs) in cisplatin-resistant hypopharyngeal cancer by comparing differentially expressed genes (DEGs) between cisplatin-resistant hypopharyngeal cancer cell lines and ordinary hypopharyngeal cancer cell lines. The authors identified a bunch of miRNAs, lncRNAs, circRNAs and mRNAs that have greater enrichment. To explore further about the related pathways, the authors demonstrate that these genes are functionally related to TNF IL-17 and JAK-STAT signaling pathway utilizing DAVID. To make the findings more convincing, the authors chose several critical miRNAs and mRNAs to perform the validation experiments.


Editorial Criteria
BASIC REPORTING
The title, abstract, introduction, methods, results and discussion are appropriate for the content of the text. Furthermore, the article is well constructed, the experiments are well conducted, and analysis is well performed. The figures are relevant, high quality, well labelled and described.
EXPERIMENTAL DESIGN
The experimental design is original and the research is within the scope of the journal.. Research question is well defined, relevant and meaningful. The methods are highly technical, ethical and logistical. Statistical methods are chosen correctly.
VALIDITY OF THE FINDINGS
All underlying data have been provided in detail. The findings are meaningful. The conclusions are well stated and relevant to the research questions.

Overall, I think this paper is novel and will be of interest to the community of hypopharyngeal cancer and cisplatin resistance in cancer in general. The statistical part is valid and makes sense. The authors make it comprehensive by integrating in vitro experiments and analysis of pathways enrichment. The main strengths of this paper is that it addresses an interesting and unexplored question, finds a novel discovery based on a carefully selected set of bioinformatic and experimental procedures. Some of the weaknesses are the not always easy readability of the text. Another limitation is that the sample size is relatively small. In general, the work is convincing except some major and minor comments below:


Major Comments:

I’m wondering if there are any ongoing clinical trials focusing on the miRNAs and mRNAs identified in this study in hypopharyngeal cancer or head and neck cancer? It will be very strong evidence for the significance of the current study if so.

I’m just wondering if there is any TCGA or GEO hypopharyngeal cancer dataset with drug resistance and transcriptome profiling data. It will be great if the results could be validated using an external dataset. Please also check if there is cell lines expression data with drug response information (eg., IC50) to be able to utilize as validation dataset, like the CCLE dataset (Cancer Cell Line Encyclopedia https://portals.broadinstitute.org/ccle ) and GDSC (Genomics of Drug Sensitivity in Cancer https://www.cancerrxgene.org/ ).


Please add the limitations of this study and future directions in the Discussion part.



Minor Comments:
Background of the Abstract: please add some background and prognosis of hypopharyngeal cancer. Then introduce cisplatin and cisplatin resistance.


It is great that a session of abbreviations was there to list all the abbreviations used in the study. I would recommend to also include abbreviations like FaDu, DEG, KEGG, ceRNA etc in that list.

All the gene names should be italic for all the gene names.

Line 30: typo for “csplatin”.

Annotated reviews are not available for download in order to protect the identity of reviewers who chose to remain anonymous.

---

## Round 0.2 · accepted · Accept

The revised manuscript has been examined by one of the two reviewers of the original submission. The changes made to the manuscript adequately address the concerns raised in the first review.

Reviewer 1 ·

Basic reporting

-

Experimental design

-

Validity of the findings

-

Additional comments

The authors addressed my points well. The manuscript can now be considered for publication!